# Small Business and Livelihood: A Study of Pashupatinath UNESCO Heritage Site of Nepal

Dipesh Kumar Ghimire [1], Prakash Gautam [2,*], Shyam Kumar Karki [1,*], Jiwnath Ghimire [3] and Isao Takagi [1]

1 Department of Economics, Soka University, Hachioji 192-8577, Japan
2 Department of Business Administration, Soka University, Hachioji 192-8577, Japan
3 Department of Community and Regional Planning, Iowa State University, Ames, IA 50011, USA
* Correspondence: akashgautam17mobile@gmail.com (P.G.); shyamkarki83@gmail.com (S.K.K.); Tel.: +81-9099790336 (P.G.); +81-8041355122 (S.K.K.)

**Abstract:** Small businesses in world heritage sites provide services to visitors and livelihood for residents. Besides the cultural and religious values promoted by these businesses, they also provide socioeconomic support to their owners. The Pashupatinath temple is known as Hindu's major religious and pilgrimage destination in South Asia. Hundreds of businesses around the temple provide services to visitors. This study evaluates the socioeconomic impacts of these small businesses around Pashupatinath temple. Using a survey of 110 businesses, binary logistic regression models find that the owners of larger businesses selling religious supplies in this area are more likely to own houses in Kathmandu and to be more satisfied with their businesses. The study also finds that businesses without permanent stalls faced severe hardship during the COVID-19 pandemic. This study assessed the socioeconomic status of a business owner through house ownership in Kathmandu, and finds that small businesses operating in the premises of the religious heritage site of Pashupatinath temple have a positive relation to the livelihood of the business owners and their families. It concludes that small-scale business in world heritage sites directly contributes to local livelihoods and economies.

**Keywords:** world heritage sites; small business; tourism; livelihood; Nepal

## 1. Introduction

There is not a single definition of small business. They differ in their capitalization levels, sales, number of employees, and incomes [1]. Nepal's Industrial Enterprise Act 2020 categorizes business sizes by fixed capital investments. According to the Act, investment ceilings for micro, small, medium, and large businesses are below 2 million, 2 to 150 million, 150 to 500 million, and above 500 million NRs, respectively (exchange rate USD 1 = Rs. 117.69, 9 September 2021) [2].

Small businesses accelerate the growth rate in low-income countries and are recognized as an important aspect of economic prosperity and people's livelihoods [3]. They are the sources of employment and income-generating activities in developing and underdeveloped countries [4]. They withstand adverse economic conditions created by supply chain problems because they depend on local products. In the meantime, they have lower capital investments and are directly associated with the creation of jobs and employment opportunities [5].

Most economies depend largely on small and medium-sized businesses. They represent around 90% of businesses and more than 50% of global job opportunities. In emerging economies, formal small and medium businesses provide up to 40% of the gross domestic product (GDP). These figures increase significantly when informal business data is incorporated [6]. Small businesses are essential to improve the livelihoods of the people. Their role

is pertinent in developing countries. Akinwale and Ogundiran (2014) identified the positive impacts of small businesses on poverty reduction [7]. Lee (2016) conducted empirical research in 316 Metropolitan Statistical Areas of USA and found that a 10 percent increase in a city's number of small businesses raises employment by 1.3 to 2.2 percent, annual payroll by 2.4 to 4.0 percent, and wage by 1.2 to 2.0 percent after ten years [8]. Duflo and Banerjee (2011) argued that small business owners could succeed and change their livelihoods but outside assistance is needed [9]. The Ministry of Finance (2016) mentions that small and medium businesses contribute 22% to the GDP and employ around 1.7 million people in Nepal [10].

Tourism creates employment opportunities and livelihood benefits for local communities [10–12] but poor people do not have access to these benefits and have to bear negative externalities [13].

Figure 1 shows the Pashupatinath temple area, and dotted paths show the survey transects for this study. The temple is located on the Bagmati riverbank in Kathmandu valley, and the area was registered on the United Nations Educational, Scientific, and Cultural Organization (UNESCO) World Heritage Sites in 1979. The construction date of the Pashupatinath temple is yet to be discovered. However, Nepal Tourism Board (NTB) mentioned that the Pashupatinath temple was rebuilt in the fifth century and renovated in 1692 A.D [14,15].

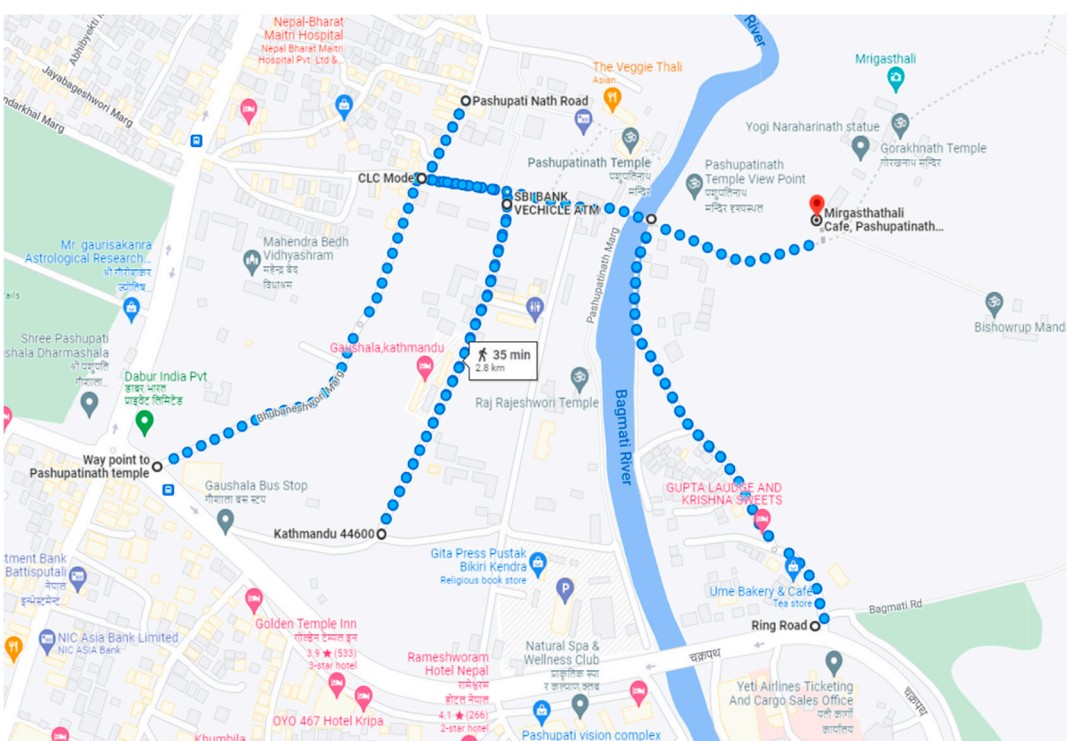

**Figure 1.** Study Area (UNESCO World Heritage Site of Pashupatinath Temple).

Pashupatinath temple is a sacred heritage site and one of the major religious and pilgrimage destinations for the Hindu religious community in Asia. Besides its significance among Hindus, the site is a tourist attraction for other religious communities, including Buddhists, Kirats, Sikha, Jains, and others [16]. Its antiquities, religious faith, mythologies, and legends attract religious and non-religious tourists from around the world.

South Asia has a rich and diverse cultural heritage, including temples, monasteries, monuments, forts, tombs, and palaces. A vibrant and ever-changing attached community surrounds each of these heritage sites. Originally, these communities were established to take care of these sites and grew as urbanization took place around these locations. There are 63 UNESCO World Heritage Sites in the continent, which are in India (40), Sri

Lanka (8), Pakistan (6), Nepal (4), Bangladesh (3), and Afghanistan (2) [17]. Fifty of the 63 sites are cultural, twelve are natural, and one is a mixed heritage site [17,18]. Among the four Nepalese sites, Kathmandu Valley (Pashupatinath, Boudhanath, Swayambhunath, Kathmandu Durbar Square, Changunarayan, Bhaktapur Durbar Square, and Patan Durbar Square) and Lumbini are cultural heritage sites; Sagarmatha National Park and Chitwan National Park are natural heritage sites [19].

The Government of Nepal (GoN) established the Pashupatinath Area Development Trust (PADT) to preserve and develop Pashupatinath temple and its surrounding areas, including *Ghaat (cremation sites), Dharmashala (cultural shelters), Pati (temporary resting areas), Sattal (gathering places), Brnhmanaal (a place where a continuous stream of water is flown through the feet of the dying person.), Bagmati Nadi (Bagmati River)* and other temples. Every structure in the territory has a historic significance. Their preservation and maintenance are the main responsibility of PADT. Originally, PADT was created by the late King Birendra in 1986 after he was exposed to different management systems of cultural heritage sites and wished to develop the same for Nepalese sites, including Pashupatinath temple [20]. It is currently operating under the PADT Act 1988 [21]. Furthermore, the GoN has announced the Pashupatinath Area Development Project as a National Pride Project [22]. However, the PADT Act and the national pride project do not include any provision regarding the protection and promotion of small businesses in this area.

Every year, many people visit the Pashupatinath premises for religious, archeological, and historical reasons. The major festivals, such as *Maha Shiva Ratri*, *Bala Chaturdashi*, and *Teej*, attract domestic and international visitors to the premises. The PADT expected more than 1.3 million people from across the world in 2021 despite the global pandemic [23].

Small businesses in the temple area have a great opportunity to sell goods and services during those festivals. Most of them are involved in curios, souvenir gifts, small hotels, coffee shops, jewelry, religious materials shops, etc. Many families' livelihoods depend on Pashupatinath's *Pooja* (a daily ritual), festivals, and funeral ceremonies.

Various studies have noted the impact of small businesses on residents' livelihoods overall. However, there is limited research on small businesses operating within religious heritage sites and their roles in the livelihoods of the surrounding residents. Pashupatinath is a famous religious heritage site and a tourist destination, although only limited academic research has been published concerning this topic.

Given the gap in the scholarship, this study focuses on education, health, and food expenses as determinants of livelihood. The main study hypothesis is that small businesses at heritage sites positively impact the availability of food, education, and health of their owner and his/her family members. Through the hypothesis evaluation, this study assesses the impact of small businesses operating in Pashupatinath premises on the livelihood of their entrepreneurs and their families, and how COVID-19 has impacted their operations and management. Shrestha (2010) argues that housing is a fixed and long-term investment, social asset, and income source in Nepal, and that it has an impact on people's livelihoods, health status, and socioeconomic progress [24]. The socioeconomic status of owners is evaluated based on their ownership of houses and other assets in Kathmandu.

## 2. Understanding Tourism, Small Business, and Livelihood

### 2.1. Operational Definitions

Tourism is "a social, cultural, and economic phenomenon which entails the movement of people to countries or places outside of their usual environment for personal or business/professional purposes. These people are called visitors (which may be either tourists or excursionists; residents or non-residents) and tourism has to do with their activities, some of which involve tourism expenditure" [25].

Religious and sacred heritage sites reflect unique cultural legacies and community traditions. Religious places have a peaceful environment that inspires awe, encourages contemplation and meditation, and instills an attitude of respect for religious values [26]. Those religious sites and various heritage sites are related to nature, culture, and architecture, and

can serve as a bridge to understanding the interconnectedness between human beings and nature [27]. In Hinduism, the mountains, rivers, hills, forests, coastlines, and landforms possess spiritual energies maintained and regulated by associated divinities described in the *purāṇas* (the ancient books) [28]. The Hindu pilgrimage, *tirtha yatra* (spiritual journey), has a religious and symbolic meaning to the believers. The protection and development of religious heritage sites greatly affect these travelers and other tourists. Moreover, religious heritage has a strong archaeological and ritual significance [29,30].

From a religious heritage conservation perspective, tourism can create financial resources for protecting natural areas and ancient sites. Tourism directly helps and rebuilds the local economy. Cultural tourism creates jobs and new business opportunities [31]. Religious and cultural heritage sites enable communities to learn about their cultural history chronologically. Furthermore, religious tourism development can improve the maintenance and preservation of these heritage sites [32,33].

The Ministry of Culture, Tourism & Civil Aviation of Nepal categorized the tourism into categories: holiday or pleasure, pilgrimage, trekking and mountaineering, and others. These tourism statistics show that the holiday or pleasure category holds the largest share of tourists, and the pilgrimage category holds the second largest in 2019 [34]. Timothy and Nyaupane (2009) considered religious and pilgrimage tourism as heritage tourism [35]. The authors also considered religious, cultural, and pilgrimage tourism as heritage tourism in this study.

Cultural heritage sites promote economic development and can play a key role in economic growth and livelihood. They contribute to sustainable development to improve the status of local communities. These sites also reflect cultural, historical, and social values, which are crucial for sustainable development [31,33,36]. Heritage tourism can be one of the most important sources of income for the local communities [37]. Greffe (2004) argued that heritage tourism can contribute to business owner's economic status and livelihoods [38]. Promotion of heritage sites, local products, and participation by inhabitants in business can reduce poverty and improve the livelihood of the owner.

### 2.2. Small Businesses and Livelihood Implications

Bolton (1971) defined small businesses based on three standards. Firstly, it has a rather tiny market share. Secondly, the proprietors operate in a personalized manner with no organized management structure. Finally, it is independent of a large enterprise [39]. Wynarczyk et al. (1993) emphasize the characteristics of small businesses [40]. According to them, there are three differences between large and small businesses. First, the uncertainty that comes with being a price taker. Second is a small consumer and product base; third, when compared to large businesses, uncertainty is related to greater objective diversities. According to Storey (1994), the concept of perfect competition does not apply to small businesses [1]. Many of them provide highly specialized services or products in isolated areas. Small businesses mobilize funds that would otherwise be idle, promoting indigenous technological knowledge. These businesses use mainly local resources and adapt easily to customer requirements. Thus, small business is always directly connected with economically poor people, and caters to the needs of the poor [41].

Small businesses and tourism reduce poverty by inspiring business development and employment creation. Cultural routes promote extensive community engagement in cultural events that develop knowledge of shared cultural history. It is a resource for innovation, creativity, smaller business formation, and the development of cultural tourist products and services [42]. Private sector comprises small and medium-sized businesses which provide services ranging from accommodation to food, transportation, and entertainment [43]; in addition, residents can take economic benefits, either through employment or by owning small businesses in heritage sites [44–46].

### 2.3. Heritage Tourism and Its Role on Livelihood

Ghimire (2019) identifies the economic impacts of small businesses associated with pilgrimage tourism in Shikoku, Japan. A total of 73.3 percent of participants in the study reported small businesses having an economic impact on their livelihoods [31]. Kusi et al. (2015) examined how micro and small businesses contribute to job creation in emerging economies, income generation, and poverty reduction [47]. Small businesses operating in historical heritage sites contribute to poverty reduction. Poverty is itself a multidimensional concept [31,48]. Education, health, and food availability determine a person's poverty level. Small businesses have a very positive effect on the livelihoods of low-income families in heritage places.

Income, employment, and financial measurement are related to livelihood [49]. The concept of livelihood is complex and interrelated with human, financial, social, physical, and natural capital. Scoones (1998) describes it as an occupation performed by people to make a livelihood in the world [50]. Majale (2002) examines that a livelihood comprises the capabilities, assets, and activities required for living [51]. Livelihood relates to people's lives and well-being; it is also an opportunity to fulfill their basic needs and requirements [48]. Bebbington (1999) and Haan and Zoomers (2005) examines that livelihood is an integrating concept of social and economic requirements of the people [52,53]. Ellis (1998) argues that livelihood comprises activities, assets, accessibilities, and processes that jointly determine the living gained by individuals or households [49]. To summarize, a livelihood is a way of meeting fundamental necessities such as food, water, housing, clothes, health, and education [54–56].

Zhao and Ritchie (2007) theorize tourism and its contribution towards improving livelihoods of poor people. They explore the "role of the poor and corresponding modes of participation in tourism". A poor person may participate as a community member and as a human resource. They also argued that tourism provides an employment opportunities and entrepreneurship to the poor [57].

Fixed asset ownership is a major determinant of the socioeconomic status of people. Due to the high price of land and housing, ownership of these in cities is more critical in defining the social and economic status of families and individuals. House ownership is a most valuable indicator of socioeconomic status [58]. Shrestha (2010) identifies that the cost of Community Housing in Kathmandu ranges from Nepalese Rupees (NPR) 5,900,000 to NPR 9,300,000 (exchange rate: USD 1 = NPR 74.45, date: 02/08/2010) per unit. Similarly, the house price to income ratio in Kathmandu is 10.6, implying that one-fourth of the Kathmandu population with a family income less than NPR 6000 cannot afford private housing [24].

Dietz & Haurin (2003) found direct and indirect positive impacts of house ownership on wealth, investment risk, children's education, and better physical and mental health [59]. Haurin et al. (2002) argue that home ownership has been linked with improving mental and physical health [60]. According to Rasmussen et al. (1997), homeowners are more likely to have access to resources that can be utilized to fund health care in later life [61]. Furthermore, Table 1 shows the relevant works as bellow.

From reviewing the literature, small businesses in heritage sites ensure income generation and economic growth, and provide job opportunities for local people. They promote the equitable distribution and productive use of local resources, which is directly associated with local livelihood improvements. The main purpose of this study is to evaluate the socioeconomic impacts of small businesses. To do so, the authors focus on house ownership, expenditure on children's education, family healthcare, and food expenses as determinants of livelihood.

**Table 1.** Works concerning local livelihoods and tourist destinations at religious heritage sites.

| Authors | Title | Key Findings |
|---|---|---|
| Keitumetse, 2011 [62] | Sustainable Development and Cultural Heritage Management | Living communities continually appropriate, reconstruct, and re-use cultural legacy to meet the current requirements. |
| Levi & Kocher, 2009 [63] | Tourism at Heritage Religious Sites | Sacred places and spiritual tourism give benefit to both tourists and communities. |
| Mydland & Grahn, 2012 [64] | Heritage values in local communities | Cultural legacy is a tool for the growth of social experiences, relationships, and exchanges. |
| Michaels, 2008 [15] | Festivals and rituals in Pashupatinath temple | Peoples worship Pashupatinath for religious merit, and festivals provide benefits to the local people. |
| Ghimire, 2019 [31] | Cross-national comparison of heritage tourism | Heritage places contribute to tourism activities and economic growth. |
| Nyaupane, 2019 [16] | Cultural heritage and tourism management | Heritage tourism has great importance in promoting sociocultural, religious, historical, and aesthetic values. It also contributes to economic growth. |
| Levi & Kocher, 2009 [63] | Tourism and heritage religious sites | Heritage religious sites are important for both tourists and local community people. |
| Kumar & Singh, 2017 [65] | Sacred Heritage City Development and Planning | Heritage has become a resource for city development. |
| Deller et al., 2018 [66] | Social capital, religion, and small business activity | There is a positive relationship between religious values, social capital, and community economic development. |
| Coffee et al., 2013 [58] | Relative residential property value as a socioeconomic status | House ownership is a most valuable indicator of socioeconomic status. |
| Zelekha et al., 2014 [67] | Religious institutions and entrepreneurship | Religious institutions have a significant impact on the tendency to become an entrepreneur. |
| Mitchell & Ashley, 2010 [68] | Tourism and Poverty Reduction | There is a strong relationship between tourism and poverty reduction. |
| Li et al., 2018 [69] | Tourism and economic growth | Tourism can contribute to reducing poverty. Economic factors such as labor, capital, and technology determine tourism efficiency and productivity. |

## 3. Methodology

### 3.1. Data Collection

The study uses a survey questionnaire [70] to understand the livelihood roles of small businesses in the world heritage site of Pashupati Temple of Nepal. Questions were focused on the duration of business ownership, economic status, household characteristics, and psychological satisfaction among business owners. Besides the survey, the study also employs key informant interviews (KIIs) to better explain the small businesses' roles in improving owners' livelihoods. Interviews allowed outreach to business owners who did not have enough time to participate in the survey.

The field survey was conducted from 5 August 2021, to 25 August 2021. The authors used snowball, convenient, and purposive sampling approaches to include and exclude survey and structured interview participants. There is no record of operating business counts in the Pashupatinath Temple area because many of them are informal and seasonal.

The sample size is determined based on the data needed for descriptive and categorical data analysis. Before deciding to include businesses in the research, the authors observed the business operations and made sure that interviews and surveys were administered among the owners of the businesses. These businesses also have many employees helping with their operations. Business employees were not included in the interviews and surveys because they may not have sufficient knowledge about the business operation and its economic status. The sample is collected from all entrepreneurs available during the survey period. A total of 110 responses were recorded during the survey period, where random and purposive snowball sampling was used to select respondents. Participation was entirely voluntary. Human subject protection was enforced throughout the study. Personal information about the respondents was not collected, including owner's name, business address, business name, and residential addresses.

### 3.2. Measurement Description

The survey questionnaire was divided into business data, characteristics of the owners, and COVID-19 impacts on businesses. Questions were asked in a 'Yes/No' form to better grasp the clear role of businesses on the economic well-being of owners and their family members. On business data, the authors asked whether the entrepreneurs were satisfied with their businesses, the duration of owners with the businesses, business size, business management training, number of employees, the volume of transactions, customers, and satisfaction with the business. On the socioeconomic characteristics of owners, the survey had questions related to the diets, family size, physical and mental health of respondents and family members, household expenditure, household income, language skills, and home ownership in the Kathmandu valley. On COVID-19 impacts, the survey included questions related to the performance of businesses during the lockdowns, the ability to perform economic obligations such as paying rents, and self and family vulnerabilities to factors such as business satisfaction, and customer ratios were assessed using five-point Likert scales [71]. The language capability of owners was measured using a multiple-choice approach. Income-related questions were used to record income levels as a continuous variable. Table 2 shows the characteristics of the participants.

**Table 2.** Survey responses (Frequencies) (N = 110).

| Characteristics | Categories | Response Percent (%) |
|---|---|---|
| Gender | Male | 60.9 |
| | Female | 39.1 |
| Age (years) | 21–30 | 9.1 |
| | 31–40 | 38.2 |
| | 41–50 | 36.4 |
| | ≥51 | 16.4 |
| Education level | Less than Secondary | 32.7 |
| | Secondary & Higher Secondary | 49.1 |
| | Bachelor's | 15.5 |
| | ≥Master's | 2.7 |
| Business Investment (Exchange rate: USD 1 = Rs. 117.69, 9 September 2021) | Less than Rs. 100 thousand | 28.2 |
| | Rs. 100 to 400 thousand | 24.5 |
| | Rs. 500 to 900 thousand | 29.1 |
| | Rs. 1 million or more | 18.2 |
| Business duration (years) | Less than 1 Year | 13.6 |
| | 1 to 4 Years | 59.1 |
| | 5 to 9 Years | 19.1 |
| | 10 Years+ | 8.2 |
| Business training recipient | Yes | 2.7 |
| | No | 97.3 |

*3.3. Data Analysis*

Given the nature of the data, descriptive statistics and a binary logistic regression model [72] were used to explore the relationship between home ownership of small business owners and livelihood characteristics. The home ownership variable was measured as binary (Yes/No). The survey question was whether the respondent owns a home in the Kathmandu valley. The regression model to test home ownership in Kathmandu was built as shown below [73]:

$$\text{logit}\{\Pr(Y = 1|x)\} = \log\left\{\frac{\Pr(Y = 1|x)}{1 - \Pr(Y = 1|x)}\right\} = \beta_0 + x'\beta \tag{1}$$

where,

Y = Home ownership variable (Yes = 1, No = 0).
x = Vector of the explanatory variables. Explanatory variables included business stall size, experience with business operation, business investment sizes, children going to private schools, family size, multilingual capacity, and monthly health expenses.
$\beta_0$ = intercept parameter.
$\beta$ = vector of slope parameters.

**4. Findings**

Table 2 shows the demographic characteristics of respondents. A total of 110 small business owners around Pashupatinath Temple responded to the survey. More than 60% of respondents were male and the majority of them (38.2%) were 31 to 40 years old. The average marriage age in Nepal is 20.7 years and the majority still prefer to live in joint families [74]. Fifty percent of the respondents had high school attendance whereas 2.7% had a master's degree. More than 59% of business owners have owned the business for 4 years or less. These businesses have low investment. More than 28% of respondents have invested less than one hundred thousand Nepali rupees, and more than 18% invested more than a million rupees. There was 18.2% who invested more than one million Nepali rupees on the business. More than 97% of respondents did not receive any formal training on formal business management training.

The small businesses in the Pashupatinath temple premises were found to positively impact the business owners' livelihoods, including education and health. One of the survey questions was to categorize businesses. Respondents were asked to categorize their businesses into three types: street/cart, small stall, and large stall, based on occupied space and nature of the operation.

Business size determines the owners' livelihood status. Businesses with smaller operations have children going to public schools and have a large amount of their spending on food. Children of small business owners are more likely to be out of school or in public schools. The small businesses operating in the Pashupatinath temple premises seem to have impacted the education of their owners' children. The impact differs according to the size of the business. The number of street/cart business owners' children not going to school is 36.11%, which is the highest among the three business categories (Table 3). Public school fees and quality of education are lower than that of private schools in Nepal. In contrast, business owners with large stalls have 84.21% of their children going to private schools, which have higher fees and a better quality education.

**Table 3.** Business Types and School Enrollment Patterns among Businesses.

| Business Type | School Type | | | |
| --- | --- | --- | --- | --- |
| | Private | Public | Out of School | Total |
| Street/Cart | 3 | 20 | 13 | 36 |
| Small Stall | 14 | 19 | 3 | 36 |
| Large Stall | 32 | 4 | 2 | 38 |

Figure 2 shows that the owners of street/cart businesses in the Pashupatinath temple area spend much of their income on food expenses. Most street/cart business owners spend 10,000–14,000 or 15,000–19,000 Nepalese Rupees (Rs) on their monthly food expenses. The situation of large businesses is different. Large-stall business owners spend Rs 20,000 to Rs 29,000 on food per month, which is the highest percentage for this category.

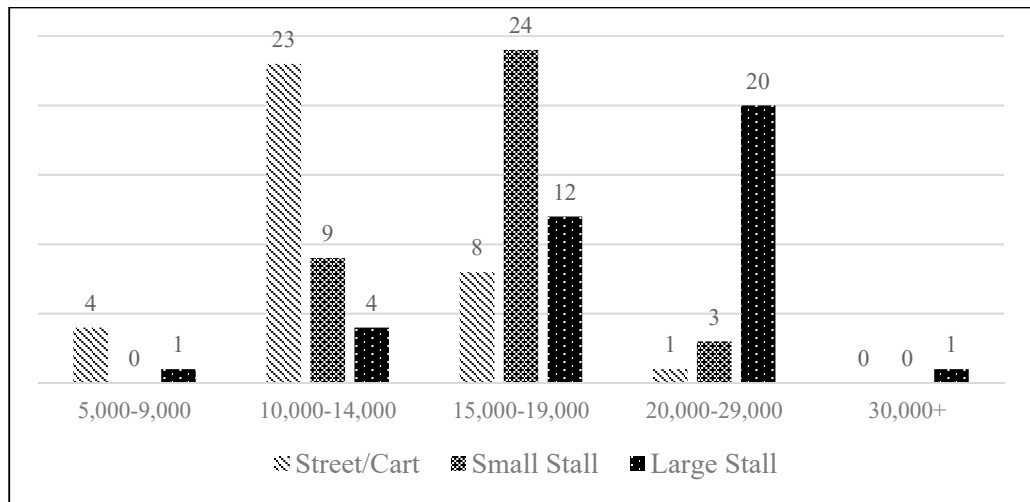

**Figure 2.** Monthly Expenditure on Food among Businesses.

Smaller businesses were identified as less vulnerable to COVID-19. Cross-tabulation was conducted to compare the pre-existing health conditions among business owners with business types to assess their sensitivity and vulnerability to COVID-19. As shown in Figure 3, 80.56% of Street/Cart businesses are without any health conditions among the owners and their household members, whereas 8.33% of small stall business owners and 15.79% of large stall owners or their family members have chronic health problems.

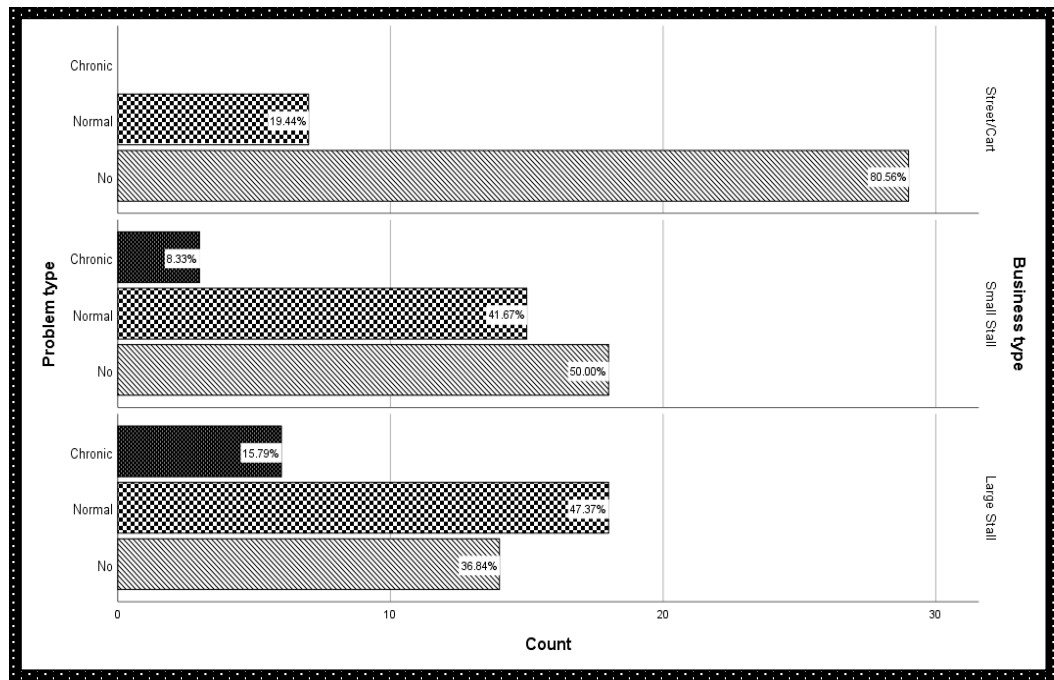

**Figure 3.** Chronic Health Conditions among Family Members Across Businesses.

Figure 4 shows the level of satisfaction among business owners. The trend shows that those who invest less than one hundred thousand Rs seem to be more dissatisfied compared to the other groups. This could be due to smaller business returns and higher market sensitivity. More than 28.18 percent with 500 thousand to 900 thousand Rs business investments are satisfied. The size of the investment determines business satisfaction. As business investment increases, the level of satisfaction of the owner also rises.

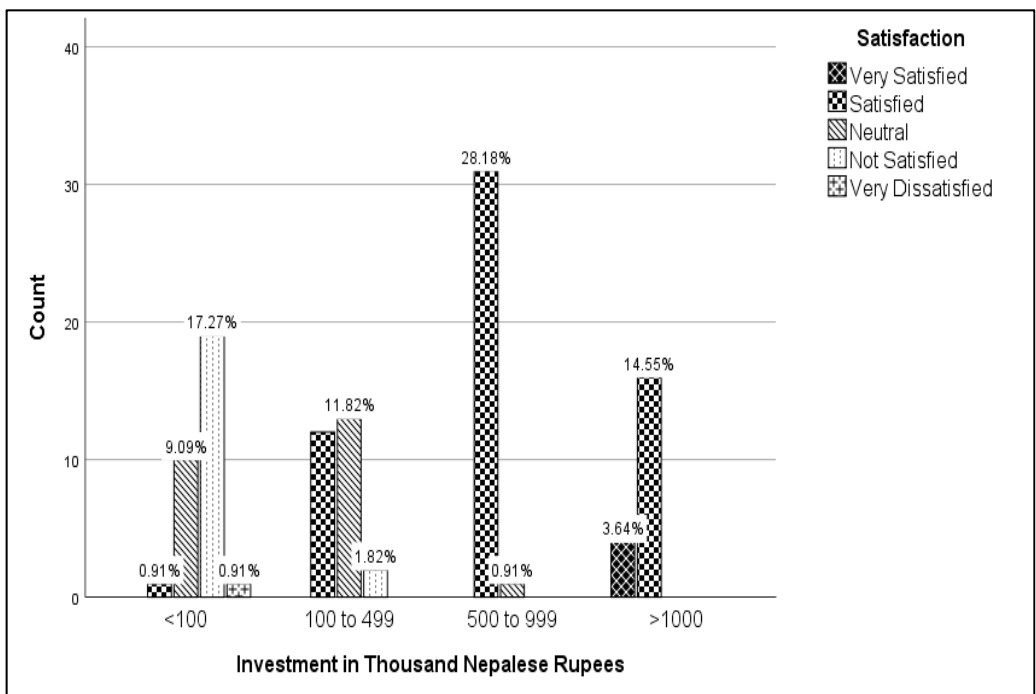

**Figure 4.** Business Satisfaction.

The binary logistic regression model was developed to evaluate the relationship between house ownership in Kathmandu and business characteristics. The model helped to better understand the role of business characteristics and lifestyle in determining house ownership among small business owners in Pashupatinath temple premises.

Table 4 shows the results from the regression model. House ownership in Kathmandu shows the economic and social status of a family. In this analysis, the house ownership in Kathmandu, among business owners in the Pashupatinath area, is treated as a dependent variable (Yes = 1, No = 0) for the binary logistic model. Business owners in the Pashupatinath Temple area with a large business stall (odds = 5.598, $p$ = 0.0489), with more than four years of business experience (odds = 6.356, $p$ = 0.0079), high school or higher education (odds = 6.751, $p$ = 0.0123), two or more working family members (odds = 7.187, $p$ = 0.0191), monthly health expenses (odds = 16.888, $p$ = 0.0191), children in public schools (odds = 6.201, $p$ = 0.0143) and multilingual capability (odds = 6.058, $p$ = 0.0246) are more likely to own house in the Kathmandu valley compared to other groups. The owners' business investments and other social characteristics, including ethnicity, gender, and age, do not significantly influence house ownership among business operators around the Pashupatinath Temple area.

House ownership influences household wealth, portfolio choice, children's education, physical and mental health, mobility, and investment capacity in developed countries [59]. In the meantime, house ownership has also been seen as a "determinant of health and quality of life" and as fulfillment of individual choices in developing countries [75]. It also has remained an indispensable phenomenon that affects every aspect of human life in developing countries [76].

**Table 4.** Logistic Regression between House Ownership and Respondents' Characteristics.

| Parameter | DF | Estimate | Standard Error | Wald Chi-Square | Odds Ratios | Pr > ChiSq |
|---|---|---|---|---|---|---|
| Intercept | 1 | −9.15 | 2.281 | 16.0916 | | <0.0001 |
| Large Business Stall | 1 | 1.7224 | 0.8745 | 3.8792 | 5.598 | 0.0489 |
| More than 4-year experience in the business | 1 | 1.8494 | 0.6965 | 7.05 | 6.356 | 0.0079 |
| Below Rs. 400,000 business investment | 1 | 0.5189 | 0.8402 | 0.3814 | 1.68 | 0.5368 |
| High school or above education | 1 | 1.9097 | 0.7629 | 6.267 | 6.751 | 0.0123 |
| Two or more working members in the family | 1 | 1.9722 | 0.8703 | 5.1356 | 7.187 | 0.0234 |
| Has monthly health expenses | 1 | 2.8266 | 1.2065 | 5.4887 | 16.888 | 0.0191 |
| Multilingual | 1 | 1.8015 | 0.8012 | 5.0551 | 6.058 | 0.0246 |
| Sends children to private schools | 1 | 1.8247 | 0.745 | 5.9982 | 6.201 | 0.0143 |

## 5. Discussion

This study aims to evaluate the socioeconomic impacts of these small businesses around Pashupatinath temple. According to our descriptive analysis, businesses in the Pashupatinath Temple area had a positive effect on their owners' livelihoods and household levels. Education, health, and food expenses seem to be managed through such businesses. However, the impact differs depending on the size of investment in the business. For example, the impact on street/cart businesses and large stall owners seems different.

Similarly, the health effects differ according to the size of the business. There is a reason behind this. Firstly, street/cart business owners may not have access to healthcare facilities. Secondly, small/cart business owners mostly live in nuclear families. Thirdly, most small/cart business owners are under 40 years of age. As a result, there is no serious chronic health problem among the small/cart business owners.

Public school education is free in Nepal. However, such schools lack quality education. On the other hand, parents must pay expensive fees in private schools. Relatively quality education is available in private school in Nepal. Authors observed that the street/cart business owners have less access to private schools due to the constraint of income. As a result, the children in this group have not gone to school or have gone only to public schools.

The household with the lowest income seems to spend the least on food [77]. The same is true of businessmen running small businesses in the Pashupatinath temple premises. Low-income entrepreneurs spend very little on food. Expenditure on food seems to vary according to the size of the business.

Our econometric results indicate that the size of the business stall, business experience, level of education, number of working families in business, and multilingual capacity, positively and significantly impact house ownership in Kathmandu. Furthermore, other social characteristics, for example, ethnicity, gender, and age, do not significantly influence house ownership among business holders around the research area.

House ownership influences household wealth, portfolio choice, children's education, physical and mental health, mobility, and investment capacity in developed countries [59]. In the meantime, house ownership has also been seen as a "determinant of health and quality of life", and as fulfillment of individual choices in developing countries [75]. It also has remained an indispensable phenomenon that affects every aspect of human life in developing countries [76].

Kiel et al. (2008) argue that the price of the house is determined by its location. Kathmandu is the capital city of Nepal [78]. Migrating to Kathmandu from the rural part of Nepal is a common phenomenon. The main reason for migrating to Kathmandu is to search for a better job, to acquire quality education, better healthcare, and to improve their livelihoods. For that reason, the population of Kathmandu is increasing every year. The increasing population leads to higher prices and higher demand for housing in Kathmandu. At the same time, having a house in Kathmandu reflects better economic status than having a house in another city. Our results revealed a positive relationship between the businesses of the Pashupatinath temple area and house ownership in Kathmandu.

A pioneering author Michael E. Porter documented that a business' size correlates with its profit [79]. This study also found the same result regarding business size and profit. We observed that the large-scale business stall has a variety of goods and services in the research area. As a result, large business stalls make more money and are more likely to have a house in Kathmandu.

According to Peters and Brijlal (2011), there is a link between the level of education of the owner/manager and the potential of the firm to develop by growing its workforce and yearly turnover [80]. Entrepreneurs with a greater degree of education and experience are likely to be more efficient in seeking, obtaining, and analyzing information regarding the availability of possibilities that lead to development [81]. Guzman and Santos (2001) argued that entrepreneurial skill is vital in overcoming survival constraints and achieving long-term success [82]. They also argued that education, professional experience, and the influence of the family, and communication skills are affecting factors of the entrepreneurial quality in their entrepreneurship theory. Our study also found similar results; those who have better education, greater years of experience in business, a greater number of family members involved in the business, and multilingual capacity positively impact house ownership in Kathmandu.

Access to quality education and healthcare facilities has been challenging for a long time in Nepal. Private schools provide quality education, albeit more expensive. The health insurance system is not common, and people have to pay all healthcare expenses themselves. Access to healthcare services is difficult for low-income households. Therefore, if the household income is good, they will pay healthcare expenses for their family members. Thus, if people do not have sufficient income, they cannot invest in family healthcare and children's quality education.

## 6. Conclusions

This study examines the impacts of small businesses on livelihoods in the Pashupatinath temple premises. The Pashupatinath temple premises is an important historical heritage and pilgrimage area in Nepal. This place is famous in South Asia as a major religious destination for pilgrims. However, the Pashupatinath area has been established not only from the point of view of religious pilgrimage but also as an important source of livelihood for many people. Pashupatinath temple premises has been established as an important place to earn a living for small business owners and their families by doing business with a small capital investment.

The descriptive and econometric analysis technique is used to analyze the primary data collected from the field survey in this study. Based on the descriptive analysis, the small business contributes to income, food consumption, health, and the education of the business holder and their family. Similarly, this study uses a binary logistic regression model to analyze the impact of small businesses on owners and their families, where the house ownership is treated as a dependent variable. Similarly, large business stalls, years of business experience, below Rs 400,000 business investment, high school or higher education, two or more working family members, monthly health expenses, children in public schools, and multilingual capability are treated as independent variables in this study. The result of this regression model found that all the independent variables besides below Rs. 400,000 business investment are statistically significant. Below Rs. 400,000 investments do not significantly impact upon house ownership among business owners. This means that the level of investment directly relates to the health, education, food expenses, sustainability, and satisfaction of the owners.

Religious and heritage sites such as Pashupatinath temple provide an important base for local people starting small businesses. These businesses positively affect the livelihood of the owner and his/her family members. Small business positively impacts the availability of food, quality education, and better healthcare facilities for the owner and their family members. As argued by Dietz et al. (2003), house ownership positively impacts on owner's wealth, children's education and better healthcare facilities, and this

study also found the same results. This study concludes that small businesses operating on the premises of Pashupatinath world heritage site have a positive relation to the livelihood of the business owners and their families.

Cultural and natural heritage is a key dimension of sustainable development. The sustainable development goals 2030 have recognized heritage and culture as parts of international development goals for the first time [17] Commercialization of these heritage sites has positive economic effects, including job creation and expansion of job opportunities [83]. Some of the activities that are commercialized in cultural, historical, and religious heritage sites are the retail sale of books, souvenirs, refreshments, admission fees, and maintaining and operating the site [83]. This research has demonstrated that religious heritage sites have crucial roles to play in the local economy through the maintenance and operation of small businesses. Future studies should document the contributions of small businesses to maintaining and operating these world heritage sites.

Although small businesses in world heritage sites are independent, their operation is heavily influenced by other activities within the premises of the site, i.e., the Pashupatinath area. Their progress and continuity are interconnected with other nearby businesses and different festivals and rituals around the site. Revenue generation of small stalls is influenced by lodging, eateries, and festivals around them. A conglomeration of diverse businesses creates revenue opportunities for all of them. This research does not consider the interconnectedness among small and medium businesses and their role in their individual revenue generation. This is a future opportunity for research.

The study does not assess how the festivals and religious rituals in these heritage sites influence the economic generations. Pashupati Temple has many popular festivals throughout the year, which attract many visitors from India, Indonesia, Europe, the Americas, and other parts of the world. Future studies will explore the role of seasonal festivals in world heritage sites on revenue generation among formal and informal businesses at these locations.

The prolonged prevalence of COVID-19 has disproportionately impacted small businesses globally. Although this study has strived to identify impacts of the pandemic on small businesses in heritage sites, findings are not conclusive because the study was conducted during the initial outbreak of the disease. Future studies should focus on long-term economic and social impacts on the economic wellbeing of these businesses.

**Author Contributions:** Conceptualization, P.G. and I.T.; Methodology, P.G., S.K.K. and J.G.; Software, P.G. and J.G.; Formal analysis, J.G.; Investigation, S.K.K. and I.T.; Data curation, D.K.G., P.G. and S.K.K.; Writing—original draft, P.G. and S.K.K.; Writing—review & editing, D.K.G., P.G., S.K.K. and J.G.; Project administration, P.G.; Funding acquisition, I.T. All authors have read and agreed to the published version of the manuscript.

**Funding:** This research received no external funding.

**Institutional Review Board Statement:** This study was approved by the Institutional Review Board for Human Research of Soka University on 23 April 2021. The approval number is 2021006.

**Informed Consent Statement:** Informed consent was obtained from all subjects involved in the study.

**Data Availability Statement:** Not applicable.

**Conflicts of Interest:** The authors declare no conflict of interest.

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
