# Peer review of "Small Business and Livelihood: A Study of Pashupatinath UNESCO Heritage Site of Nepal"

_sustainability, doi:10.3390/su15010612_

Round 1
Reviewer 1 Report
Here are some comments or the guidelines how to improve this manuscript.
1) Most economies depend largely on small and medium-sized businesses. They represent for around 90% of businesses and more than 50% of global job opportunities. In emerging economies, formal small and medium businesses provide up to 40% of national (GDP).
-Do you have any information of this type related to Nepal or its capital? It would be good to include it.
2) The socioeconomic status of owners is evaluated based on their ownership of houses and other assets in Kathmandu.
-Again, do you have any information of this type related to Nepal or its capital? It would be good to include it.
3) Due to high land and house prices, their ownership in cities is more critical to define social and economic status of families and individuals.
-What are those prices?
4) I am bit surprised that the authors did not include any hypotheses and present the results through a conceptual graphical model. That would be more convenient for all readers.
5) What are the study limitations, implications and suggestions for a follow-up research?
Author Response
We have incorporated all the comments; we would like to request you find the attachment.

Reviewer 2 Report
Small business and livelihood need to be strengthened in literature review as subsection
Method has to become data collection, measurement description, data analysis, and research model using subsections
Authors also need to why the destination is worthy to inspect related to livelihood and small.business in intrduction section.
Moreover, ,research hypotheses need to be proposed based on the econometric.model.
Conclusion parts should emphasize the theoretical implication of this work.
Author Response

(The authors gave the same response as above.)

Reviewer 3 Report
The article describes an important problem related to raising the standard of living in a developing region thanks to economic activity in the area of ​​a unique tourist destination.
- The introduction should include information on the purpose of the research (main objective and possibly specific objectives), information on research methods, research procedure, expected results in the form of a contribution to the theory of science and practice of tourism business;
- Lines 127-128 - wrong definition of tourism, "Tourism is a travel for a limited period for leisure, religious, family, business and other purposes.", I propose to quote the definition the World Tourism Organisation (www.unwto.org);
- What form of tourism do the Authors describe: cultural, religious or pilgrimage? Provide definitions and indicate the dominant form of tourism in the destination;
- The review of the literature gives the impression of disorder, I suggest dividing the text into two or three subchapters, concerning the definition of enterprises and tourist enterprises, and the types of tourism and finally the characteristics of the researched area;
- Lines 214-216 include (probably) theses/hypotheses "The authors focus on education, health, and food expenses as determinants of livelihood. Small business has a positive impact on the availability of food, education and health of its owner and his/her family members.” I propose to formulate hypothesis;
- Moreover, most of the respondents are relatively young (between 30 and 40 years old), so many of them may not have family yet;
- How was the research sample selected? What is the population size? Please complete this information in the chapter 3;
- The sample must be shown to be representative;
- On what basis did the authors conclude that "The small business in the Pashupatinath temple premises was found to have a positive impact on the business owners' livelihoods, including on education and health."? Or maybe it's the other way - educated and healthy people start a business? - Almost 74% of enterprises were founded in the last four years, it is assumed that the reason for this sytuation were political or economic factors, which were omitted in the article.
- The presented research results give a lot to think about, because they show a trend opposite to the global one, where many enterprises were liquidated during the pandemic, while the presented results show the opposite trend. It is therefore necessary to explain what the phenomenon of the described case is. Is it due to the greater number of pilgrims due to the fear of a pandemic, or other factors?
- It would be very interesting to present the results regarding the size of enterprises, job satisfaction and the assessment of children's health and education, taking into account the gender factor of the company owner;
- There are no references to tourism in the discussion and conclusions;
- The Conclusions should include input to the theory of science and practice and recommendations to the further research.
In conclusion, the topic discussed in the article is interesting and important, but I recommend completing the text before publishing.
Author Response

(The authors gave the same response as above.)

Reviewer 4 Report
This research focuses on an important aspect of tourism sustainability, that is, the contribution of the tourism sector towards economic development of local communities. It is an interesting case study that is used to explore interconnections between tourism activity and economic well-being of small business owners, e.g. through householding.
The article is very well presented, with a good literature review, suitable methods and pertinent discussion. I raise only one question that could potentially be relevant: did the authors consider collaboration between small businesses in the area as a significant variable? Although small stall owners have independent activity, when taking the area surrounding the temple as a whole, it would seem likely that there would be some interconnectedness between certain types of businesses, for example, food stalls, small lodgings or souvenir shops, which in turn could increase chances of higher revenue for partners. Was this variable considered in the study but found to be irrelevant in the data, or was it not all at considered by the authors? If not, maybe it could be included as something for future research.
Another minor issue, in some sections (particularly section 1) the English is poor, so I would consider getting professional editing for English.
Author Response

(The authors gave the same response as above.)

Round 2
Reviewer 2 Report
The revision is done well.